# Role of Biomarkers for the Diagnosis of Prion Diseases: A Narrative Review

**DOI:** 10.3390/medicina58040473

**Published:** 2022-03-25

**Authors:** Miren Altuna, Iñigo Ruiz, María Victoria Zelaya, Maite Mendioroz

**Affiliations:** 1Sant Pau Memory Unit, Hospital de la Santa Creu i Sant Pau—Biomedical Research Institute Sant Pau—Universitat Autònoma de Barcelona, 08041 Barcelona, Spain; iruizb@santpau.cat; 2Centre of Biomedical Investigation Network for Neurodegenerative Diseases (CIBERNED), 28031 Madrid, Spain; 3CITA-Alzheimer Foundation, 20009 Donostia-San Sebastián, Spain; 4Department of Pathological Anatomy, Hospital Universitario de Navarra, 31008 Pamplona, Spain; mv.zelaya.huerta@navarra.es; 5Department of Neurology, Hospital Universitario de Navarra, 31008 Pamplona, Spain; maitemendilab@gmail.com; 6Neuroepigenetics Laboratory-Navarrabiomed, Hospital Universitario de Navarra, Universidad Pública de Navarra (UPNA), IdiSNA (Navarra Institute for Health Research), 31006 Pamplona, Spain

**Keywords:** prion disease, biomarkers, diagnosis, dementia, neurodegeneration

## Abstract

Prion diseases are progressive and irreversible neurodegenerative disorders with a low incidence (1.5–2 cases per million per year). Genetic (10–15%), acquired (anecdotal) and sporadic (85%) forms of the disease have been described. The clinical spectrum of prion diseases is very varied, although the most common symptoms are rapidly progressive dementia, cerebellar ataxia and myoclonus. Mean life expectancy from the onset of symptoms is 6 months. There are currently diagnostic criteria based on clinical phenotype, as well as neuroimaging biomarkers (magnetic resonance imaging), neurophysiological tests (electroencephalogram and polysomnogram), and cerebrospinal fluid biomarkers (14-3-3 protein and real-time quaking-induced conversion (RT-QuIC)). The sensitivity and specificity of some of these tests (electroencephalogram and 14-3-3 protein) is under debate and the applicability of other tests, such as RT-QuIC, is not universal. However, the usefulness of these biomarkers beyond the most frequent prion disease, sporadic Creutzfeldt–Jakob disease, remains unclear. Therefore, research is being carried out on new, more efficient cerebrospinal fluid biomarkers (total tau, ratio total tau/phosphorylated tau and neurofilament light chain) and potential blood biomarkers (neurofilament light chain, among others) to try to universalize access to early diagnosis in the case of prion diseases.

## 1. Introduction

Prion diseases, also known as transmissible spongiform encephalopathies, are rapidly progressive and irremediably fatal neurodegenerative disorders. The average life expectancy is six months, but great variability in the duration of the clinical course of the disease has been described, lasting from weeks to years [1,2,3]. The cause is the aggregation of a misfolded prion protein scrapie (PrPSc). PrPSc is the abnormal conformational isoform of the normal cellular prion protein (PrPc) located on the cell surface of central nervous system neurons, the exact function of which remains unknown [4]. PrPSc is able to propagate and to aggregate in the brain tissue. PrPsc is neurotoxic and its accumulation leads to synaptic degeneration and disorganization, which induces neuronal loss and spongiform changes. Indeed, a reduction of over 30% in the relative synaptic index has been reported in prion disease-affected brains [1,3,5].

Prion diseases are classified into sporadic (85%), genetic (10–15%) (due to mutations in the prion protein gene (*PRNP*)) and acquired (exceptional) forms (Figure 1). Prion diseases show a huge variety of cognitive, motor and neuropsychiatric symptoms. Almost 90% of the sporadic cases are due to sporadic Creutzfeldt–Jakob disease (sCJD) with an incidence close to 1.5–2 cases per million persons per year [3]. In addition to familial Creutzfeldt–Jakob disease, genetic causes include familial fatal insomnia (FFI) (of which there is also a very rare sporadic form), Gerstmann–Straüssler–Scheincker (GSS) and Huntington-disease-like 1 (HDL1) [6,7]. In 1996, variant Creutzfeldt–Jakob (vCJD), a zoonotic prion disease acquired by consumption of cattle contaminated by bovine spongiform encephalopathy, was described for the first time in the United Kingdom [8,9,10]. Fortunately, sanitary measures and food chain regulation have contributed to the near disappearance of vCJD [8,9]. Regardless of the type of prionopathy, early diagnosis is a challenge due to the great phenotypic variability, with the most frequent symptoms being rapidly progressive dementia, cerebellar ataxia and myoclonus [3].

Currently, the definitive diagnosis of prion diseases requires postmortem anatomopathological examination or brain biopsy, which is not feasible in clinical practice [1,3] (Table 1). Postmortem neuropathological study is also necessary to classify cases into the different subtypes of sCJD. Different clinicopathological subtypes of sCJD, defined by methionine/valine polymorphism at codon 129 of the *PRNP* gene and the type (based on the size of protease-resistant fragments) of PrPSc accumulated in the brain, have been described [3,14]. Despite the development of new diagnostic tools, postmortem anatomopathological study is the gold standard technique to confirm, or completely rule out, the diagnosis of a prion disease and, therefore, classifications that relate clinical and anatomical changes are still relevant. The molecular subtype of sCJD is an important prognostic marker for patient survival [3].

In recent decades, efforts have been made to advance the use of biomarkers to allow an early diagnosis of the different prionopathies. Neurophysiological and neuroimaging biomarkers and different cerebrospinal fluid (CSF) analytes (14-3-3 protein and total tau) have been progressively incorporated into the diagnostic criteria. Special mention should be made of the real-time quaking-induced conversion (RT-QuIC) technique for the detection of prion protein in CSF, a technique with the most promising results for the accurate premortem diagnosis of sCJD and other prionopathies, which is already incorporated into the latest diagnostic criteria [3] (Table 1). Despite its promising results, the RT-QuIC technique is not universally available; therefore, research into other more accessible biomarkers with potentially high diagnostic performance has continued over the last few years.

This narrative review aims to examine the currently accepted neurophysiological, neuroimaging and cerebrospinal fluid (CSF) biomarkers for diagnosis and to investigate alternative CSF and peripheral blood biomarkers that have been recently proposed (Figure 2). We highlight those biomarkers that are more easily accessible, including blood biomarkers, which would truly represent a diagnostic revolution in prionopathies. The existing differences in biomarker performance between different prionopathies will also be emphasized, although it is true that most of the work reviewed has focused on sCJD. The examination, in combination, of current diagnostic criteria and of multimodal diagnostic biomarkers (e.g., neurophysiological, neuroimaging, genetic, CSF and plasma analytes), available both for use in clinical practice and in the development phase, not only limited to CJD but also to the rest of the prionopathies, is considered of special interest in this review.

## 2. Neurophysiological Biomarkers

### 2.1. Electroencephalogram (EEG)

Periodic sharp-wave complexes (PSWCs), generalized and/or lateralized complexes, characterized by strictly periodic cerebral potentials at a frequency of 1 Hz, are the typical EEG finding commonly associated with sCJD. PSWCs tend to disappear during sleep and may be attenuated by some psychotropic drugs [15]. PSWCs are observed in 67–95% of patients with sCJD, with a sensitivity of 65% and specificity of 90% [16,17]. In turn, the main background frequency and the α/θ power ratio in quantitative EEG seems to be related to clinical progression and has been suggested as a useful tool for follow-up monitoring in prion diseases [18]. EEG could also be useful for the detection of non-convulsive status epilepticus, which, although infrequent, has an increased risk in some prionopathies, such as sCJD [3].

Advantages: EEG is an economically accessible and safe technique. It is available in most healthcare centers and can be performed repeatedly during clinical follow-up if required.

Limitations: The diagnosis performance of EEG improves with the clinical course of the disease. Therefore, it is a biomarker that is mainly related to the late symptomatic phases of sCJD [13,15]. In early stages of the disease, unspecific alterations, such as diffuse slowing and frontal rhythmic delta activity (FIRDA), are more frequent [15]. Among the other limitations, it should be noted that similar alterations have been described in other neurodegenerative dementias, such as Alzheimer’s disease (AD) and Lewy body dementia (LBD), although less frequently. It is also noteworthy that the probability of detection of PSWC is much lower in other prionopathies, such as FFI or GSS, and even in the MV2, VV2 and MM2 forms of sCJD [3,17,19].

### 2.2. Polysomnogram (PSG)

In cases of familial or sporadic fatal insomnia the demonstration of an early and progressive reduction in total sleep time, the loss of sleep spindles and K-complexes, the disruption of normal sleep structure, sleep fragmentation, and periods of subwakefulness interrupted by brief episodes of REM sleep, with or without atonia, often associated with dream enactment behavior, is a diagnostic criterion [20]. In the case of sCJD, sleep anomalies are not recognized as a diagnostic criterion, although it is common to detect loss of normal sleep EEG architecture, sleep-disordered breathing [21] and periodic leg movement disorders in the PSG [22].

Advantages: A non-invasive technique with high diagnostic performance for both sporadic and familial fatal insomnia.

Limitations: Not universally accessible and not useful for differential diagnosis of the most frequent causes of rapidly progressive neurodegenerative dementias apart from fatal insomnia.

## 3. Neuroimaging Biomarkers

### 3.1. Brain Magnetic Resonance (MR)

The presence of hyperintensities in T2 and fluid-attenuated inversion recovery imaging (FLAIR) sequences are frequent in the MR images of sCJD specially involving basal ganglia [16], as well as the restriction of diffusion (DWI) in at least two cortical regions (ribboning) or/and restricted diffusion predominantly in the caudate nucleus, putamen and/or thalamus [3,17,23,24]. However, these typical signs on MR images are not pathognomonic of sCJD and could also be induced, though rarely, by toxic metabolic encephalopathies, progressive multifocal dementia, autoimmune encephalitis, CNS lymphoma, vasculitis and infectious etiologies [24].

The MR pattern could also be useful, mainly when combined with *PRNP* polymorphisms, to differentiate sCJD molecular subtypes [17]. In the MM1 subtype of sCJD, the caudate nucleus has unilateral or bilateral asymmetrical involvement, and the involvement of the thalamus is more frequent in VV2 and MV2 subtypes [25]. The presence of a pulvinar sign (high signal on FLAIR and DWI) is highly suggestive of vCJD [3,26]. Moreover, altered diffusion in the striatum, thalamus and frontal and occipital cortices has been reported in GSS [6]. The sensitivity of sCJD diagnosis using MR varies according to different studies between 80 and 92%; the same is true for specificity, with a range between 74 and 98% [3,27,28]. Recently, the use of MR spectroscopy to determine the N-acetylaspartate (NAA)/creatine (Cr) ratio has been suggested as a useful parameter for predicting the clinical course in sCJD, as lower NAA/Cr is related to shorter disease duration [29].

Advantages: Structural neuroimaging is a mandatory test for the differential diagnosis of cognitive impairment, and, in the case of rapidly progressive dementias, the MR evaluation is crucial. Changes in restriction occur early in the context of sCJD [3,30]. In contrast with EEG, MR diffusion abnormalities are an early phenomenon, being detectable at least one year before the onset of symptoms in asymptomatic *PRNP* mutation carriers [31], making it useful for early diagnosis.

Limitations: At the time to perform the MR, a clinical suspicion of possible prion disease must be reported because is advisable to perform an MR evaluation with a specific protocol including diffusion sequences (DWI/ADC).

### 3.2. Fluorodesoxyglucose Positron Emission Tomography (PET-FDG)

Decreased glucose metabolism in the neocortex affecting extensive cortical regions (frontal, parietal and occipital cortices) and basal ganglia has been reported in sCJD but does not seem to be useful for differential diagnosis with other neurodegenerative dementias [28]. However, the hypometabolism of medial temporal area seems to be significantly less frequent compared to other neurodegenerative dementias [32]. Instead, it may be useful for the diagnosis of infrequent forms of sporadic fatal insomnia, where hypometabolism in the thalamic region is early and characteristic [3,6].

Advantages: It could be useful for helping with the diagnosis of sporadic fatal insomnia.

Limitations: It is an expensive test that is not usually performed in cognitive decline screening. There are no specific hypometabolism patterns that could be useful for diagnosing sCJD or performing a differential diagnosis with other prionopathies.

## 4. Genetics

*PRNP* gene sequencing is the primary diagnostic technique in genetic prion disease. *PRNP* mutations account for 10–15% of all human prion syndromes [3]. The detection of *PRNP* gene mutations can be performed by sequencing DNA from patient blood specimens or a decedent’s unfixed autopsy tissue. All the genetic forms of prion disease are linked to *PRNP* mutations and include point mutations, octapeptide repeat insertions and deletions. Many different mutations have been linked to genetic CJD, although the most common worldwide is E200K [2]. The penetrance of *PRNP* mutations is assumed to be close to 100% although real-life data are lacking. In the specific case of the E200K mutation, the most widespread significant variability has been detected with penetrance ranging from 60 to 90% among different populations [2]. The D178N mutation is present in all families with FFI. In individuals of European ancestry, five variants account for up to 85% of the pathogenic *PRNP* variants (Table 2). Therefore, the first step in genetic analysis is to determine whether these variants exist, and, if not, the entire gene should be sequenced [2]. However, in specific geographic regions with higher prevalence of gCJD, an adaptation of the preliminary genetic analysis could be advisable.

In addition, the polymorphism of codon 129 of the *PRNP* gene has potential relevance as it may influence susceptibility to both variant and sporadic forms of CJD: 85–95% of sCJD cases are methionine homozygous at codon 129, compared to 49% in healthy controls [2,17]. However, the codon 129 polymorphism has primarily been investigated in research studies and is not currently used in the diagnostic work-up of prion disease. Interestingly, stratifying patients by codon 129 polymorphism could have a possible role in future clinical trials [17].

## 5. Cerebrospinal Fluid (CSF) Biomarkers

### 5.1. Biochemical Analysis

No differences in the quantity of proteins, glucose concentration and total cell number assessed in CSF are detected in prion diseases compared to controls [33].

Advantages: The existence of significant biochemical alterations in CSF may help to check for other etiologies, e.g., inflammatory.

Limitations: Lumbar puncture is considered an invasive test, even though it is currently performed in clinical practice for the early diagnosis of neurodegenerative disorders, such as AD.

### 5.2. CSF Surrogate Biomarkers

#### 5.2.1. 14-3-3 Protein

The gamma-isoform of the 14-3-3 protein (14-3-3 gamma) expressed in neurons could be a specific marker for neuronal damage and is useful for sCJD diagnosis [34,35,36]. The pathological mechanisms leading to the accumulation of 14-3-3 protein in CSF are not fully understood; however, neuronal loss followed by cell lysis is assumed to cause increase in 14-3-3 levels [37]. Currently, the detection of increased 14-3-3 protein in CSF is used as a molecular diagnostic criterion for patients that are clinically compatible with sCJD [37]. The diagnosis performance is lower for genetic forms of prion diseases and for sporadic fatal insomnia [38,39]. The diagnostic performance of quantitative enzyme-linked immunosorbent (ELISA) assay for 14-3-3 is higher in comparison with western blotting (WB) [34,40]. Combination with other surrogate biomarkers, such as the determination of total tau (t-tau) and the ratio of t-tau/phosphorylated tau (p-tau), significantly increases the specificity [14,36,41,42,43].

Advantages: It has high sensitivity (86–97%) for sCJD [44,45,46] (Figure 3). Combination with t-tau and the ratio of t-tau/p-tau has very good sensitivity and specificity and is more accessible compared to other techniques, such as RT-QuIC.

Limitations: Protein 14-3-3 is not as specific as was initially thought [16]; specificity could vary between 75.6 and 91% for sCJD [42]. Acute neurological conditions, such as stroke, status epilepticus or inflammatory encephalopathies, can also increase 14-3-3 protein levels [47]. In turn, the approach shows poor performance in the diagnosis of infrequent prion diseases.

#### 5.2.2. Total Tau (t-tau) and Total Tau/Phosphorylated Tau (t-tau/p-tau) Ratio

Elevation of CSF t-tau levels is correlated with axonal neurodegeneration rate in many different neurological conditions, while p-tau is increased in AD but not in other neurodegenerative disorders. The determination of levels of t-tau and the ratio of t-tau/p-tau could be useful for the diagnosis of CJD, preferably combined with other diagnostic tools [16,48]. Sensitivity and specificity vary depending on the cut-off point established for both t-tau and t-tau/p-tau ratio values but very high sensitivity (85%) and specificity (98.6%) can be achieved, both of which are higher compared to the 14-3-3 protein performance [16,49,50,51,52,53]. In turn, the most highly elevated levels of t-tau are observed in the MM1, MV1 and VV2 types with classical symptomatology [54]. Therefore, it has been suggested that t-tau can be used in the diagnostic assessment of prion protein type when the codon 129 genotype is known and could provide valuable information for physicians about the prognosis [55]. Increase in t-tau and t-tau/p-tau is related to a shortened life expectancy, which could be explained because t-tau reflects neuronal damage [1,48,54,56] and t-tau levels continue to increase during the progression of the disease [57]. Levels of t-tau correlate with disease burden as assessed by cortical involvement evaluated by DWI sequence of MR [58].

Advantages: High specificity for t-tau and t-tau/p-tau ratio for sCJD. Diagnostic performance improves for both genetic and sporadic CJD if combined with 14-3-3 protein [43,59,60]

Limitations: t-tau could be increased in other neurodegenerative conditions but not as much as in prionopathies.

#### 5.2.3. Neurofilament Light Chain Protein (NfL)

NfL is a neuronal cytoskeleton component and is released when there is neuronal damage in a wide range of conditions, making this a very good biomarker of neurodegeneration [61,62]. NfL is increased in all sCJD subtypes, including those which typically show low values of t-tau and negative protein 14-3-3 (e.g., sCJD MV2K, MM2C and gCJD E200K) [5,63]. It has outstanding sensitivity to detect sCJD, higher than 95%, but a very low specificity of 43.1% [14,64].

Limitations: Low specificity makes it less useful compared to the determination of t-tau or the combination of t-tau or ratio t-tau/p-tau + 14-3-3 protein in the differential diagnosis of rapidly progressive dementias [14].

Advantages: Due to high sensitivity, this could be useful combined with other surrogate biomarkers for the early diagnosis of prionopathies.

#### 5.2.4. Other Biomarkers

**Alpha synuclein**, a neuronal protein especially abundant at presynaptic regions, stands out among other promising surrogate biomarkers. Its levels are increased in both genetic and sporadic forms of CJD, with good diagnostic performance (sensitivity 98% and specificity 97%); in addition, there is an inverse correlation between alpha synuclein levels and disease duration in CJD [61,65,66,67,68]. In turn, **neurogranin**, related to synaptic plasticity, has been shown to be increased in sCJD compared to controls and other neurodegenerative dementias with a diagnostic yield similar to 14-3-3 protein in the early stages of the disease, without significant variations with disease progression [5,10]. It has been speculated that neurogranin levels could be useful to differentiate between different subtypes of CJD (different concentrations having been reported according to the clinicopathological subtypes) [5]. Interestingly, it has been reported that **ubiquitin**, which marks neuritic damage, dysfunctional protostasis and neuroinflammation, has higher concentrations in CJD than in controls and other neurodegenerative dementias, and especially in less frequent forms of sCJD, such as MM(V)1 [69,70]. One study has also shown higher levels of **calmodulin**, a ubiquitous calcium-binding protein, in sCJD compared to other neurodegenerative dementias, particularly in those with higher levels of t-tau [71].

Markers related to oxidative stress have also been postulated, as is the case for **mitochondrial malate dehydrogenase 1 (MDH1)**. Increase in MDH1 would have a sensitivity of 97.5% and specificity of 95.6% for the diagnosis of sCJD with positive correlation for t-tau and 14-3-3 protein concentrations according to preliminary data from several studies with small or very heterogeneous samples [72,73]; however, this must be confirmed in future studies prior to its future possible use in clinical practice.

A single study indicates that **glial biomarkers,** such as YKL-40, CHIT-1 and GFAP, are significantly increased in the VV2 form of CJD compared to other neurodegenerative dementias and are positively correlated with symptomatic progression of the disease [10,74]. An increase in these biomarkers has even been observed in presymptomatic cases of GSS [74].

Biomarkers related to iron metabolism, such as **transferrin**, have also been studied. A single study has postulated that elevated transferrin could be used in combination with t-tau to increase its diagnostic performance [75].

Advantages: The discovery of new biomarkers related to different pathophysiological processes can help us to better understand the disease itself and thus identify possible therapeutic targets.

Limitations: At present, none of these biomarkers are approved for use, either in isolation, or in combination with core biomarkers, in neurogenerative diseases. Further studies are needed to confirm the benefit of their application in clinical practice.

### 5.3. CSF Prion Proteins

#### 5.3.1. Total PrP

Total PrP levels in the CSF of patients with prion disease tend to be reduced compared to controls. It has been speculated that this may result from the sequestration of soluble monomeric protein into aggregates in the brain (analogous to the proposed mechanism for reduction of CSF Aβ1-42 in AD) [65,76,77]. Specificity for diagnosing prionopathies of reduced total PrP in CSF seems to be moderate and unlikely to be used for differential diagnosis with other causes of rapidly progressive dementia, at least if not combined with surrogate CSF biomarkers, such as 14-3-3 protein and t-tau or ratio of t-tau/p-tau [37,61,78]. On the other hand, it has been suggested that it could have potential applicability for monitoring response to future disease-modifying treatments because there are significant differences in genetic forms of CJD between *PRNP* mutation carriers and non-carriers years before symptom onset, and levels of test-retest of total PrP are stable during follow-up [79,80].

Advantages: This biomarker directly reflects the pathophysiology of prion diseases and therefore has a potential role in the follow-up of possible future treatments.

Limitations: It has suboptimal performance, no better than that described for surrogate biomarkers with applications in clinical practice. Therefore, it is unlikely to be indicated for use as a diagnostic biomarker in the future.

#### 5.3.2. Prion Real-Time Quaking-Induced Conversion (RT-QuIC)

Aside from brain biopsy, RT-QuIC, an ultrasensitive in vitro PrPSc amplification assay (Figure 4), is the only disease-specific antemortem diagnostic biomarker that directly detects the pathological prion protein (PrPSc) and since 2018 has been incorporated in the current diagnostic criteria [17,41,42,62,81,82,83]. A very high sensitivity (80–96%) and virtually full specificity (99–100%) has been reported for RT-QuIC for sCJD [13] (Figure 3). However, the sensitivity is lower in MM1, the most frequent subtype of sCJD [41], and is even lower in genetic forms of CJD, GSS and FFI, as well as in sporadic fatal insomnia. The most plausible explanation for this finding is strain variability [76,81]. Nevertheless, the RT-QuIC performance continues to be higher compared to 14-3-3 protein and to the combination of 14-3-3 protein and t-tau or t-tau/p-tau ratio for genetic forms of CJD, GSS and FFI [77]. However, it has not demonstrated an ability to differentiate between different subtypes of sCJD [84]. In turn, RT-QuIC presents increased costs and less inter-laboratory standardization data is available compared to the use of approved surrogate biomarkers (14-3-3 protein and t-tau) [85].

Advantages: The ability to detect prion protein with high sensitivity and specificity of almost 100%. Included as a diagnostic criterion.

Limitations: Technically more complex than the determination of other surrogate biomarkers that are measurable in CSF (14-3-3 protein and total tau), and accessible in fewer hospitals. Poorer performance for atypical variants of sCJD and genetic forms of CJD, GSS and familial and/or sporadic fatal insomnia.

## 6. Plasma Biomarkers

### 6.1. NfL

Increased levels of plasma NfL have been reported in CJD, significantly higher than in other neurodegenerative dementias (e.g., AD, LBD and frontotemporal dementia -FTD-) and obviously compared to controls. One study has shown an AUC of 0.93 to discriminate CJD from non-CJD dementias [87] and another has reported a sensitivity of 100% and specificity of 85.5% for the diagnosis of different prionopathies (sporadic, genetic and iatrogenic CJD and GSS) [61,88]. However, it has been suggested that performance of NfL is better in CSF compared to plasma [89]. The increase in plasma NfL occurs in the early stages of the disease, before symptom onset, and it seems to be always altered if the prion conversion assay is positive in CSF. The levels of NfL continue to increase with the symptomatic progression of the disease [88,90,91].

Advantages: An accessible biomarker with a huge number of studies suggesting its utility for discriminating neurodegenerative vs. non-neurodegenerative dementias. Altered from the early stages of disease, so therefore useful for early diagnosis.

Limitations: Not a specific biomarker of prion diseases; also increased in other neurodegenerative dementias, although not to the same extent.

### 6.2. t-tau

Plasma t-tau levels are higher in CJD, in sporadic, iatrogenic and genetic forms, compared not only to controls, but also significantly when compared to other neurodegenerative dementias [25,39,41]. t-tau is particularly increased in sCJD patients who are homozygous for methionine at codon 129 of the *PRNP* gene [92]. Plasma t-tau seems to have a moderate association with disease duration, offering a moderate survival prediction capacity [93]. One study has reported a sensitivity of 84.6% and specificity of 96.2% [88], but it seems that both sensitivity and specificity are lower in general for plasma t-tau compared to CSF t-tau [89].

Advantages: This could be useful to discriminate from non-neurodegenerative causes of cognitive decline.

Limitations: It is a non-specific biomarker of neurodegeneration that does not clearly improve performance compared to use of plasma NfL.

### 6.3. YKL-40

Increased levels of plasma YKL-40 have been detected in different sporadic and genetic forms of prionopathies compared to controls. Higher levels are detected in later stages of the disease [94]. Due to the limited capacity to discriminate from other neurodegenerative dementias and moderate ability to distinguish from healthy controls, YKL-40 does not seem to be a good diagnostic biomarker but could have a role in disease monitoring [94].

Advantages: Could be a potential plasma biomarker related to progression of the disease.

Limitations: Very limited capacity to discriminate prion diseases from other neurodegenerative diseases; doubtful utility as an additional biomarker for diagnosis.

### 6.4. MicroRNA

The blood microRNA profile has been suggested as a complementary test to use, together with plasma neurodegeneration biomarkers. One study, with a small sample size, has shown different microRNA expression in sCJD compared not only to controls but also to AD [95].

Advantages: Could be another non-invasive biomarker of sCJD.

Limitations: Very limited data; does not seem to be applicable in clinical practice, at least in the short term.

### 6.5. Total Prion Protein (t-PrP)

The existence of vascular pathology with loss of integrity of the blood brain barrier (BBB) has been described in more than 40% of brains with prion disease. With loss of BBB integrity, it is expected that dying endothelial cells in the intracranial capillary vascular system, and specially dying neurons, can release PrP to the blood circulation system. Increased levels of plasma t-PrP have been reported in sporadic, genetic and variant CJD. However, increase in plasma t-PrP is not specific to prion diseases because it has also been reported in other conditions, such as AD, FTD, or LBD [96]. Therefore, it seems to be useful to discriminate between neurodegenerative and non-neurodegenerative dementias, but less clearly between prion diseases and other neurodegenerative disorders, even though higher levels have been reported in classical sCJD compared to other neurodegenerative dementias [96]. Interestingly, there is a dissociation between t-PrP CSF and plasma levels in sCJD, which are increased in plasma but decreased in CSF [62,96]. The presence of higher t-PrP levels in sCJD cases harboring MM and VV at PRNP codon 129 compared to MV carriers has also been reported, suggesting a better performance for homozygous carriers [88]. A mild association has been reported between CSF markers of neuronal injury (14-3-3 protein and NfL) and plasmatic t-PrP, suggesting that plasma t-PrP might be regarded as a biomarker of neurodegeneration, in contrast to CSF t-PrP, which probably reflects pathogenic PrP aggregation occurring in prion disease patients. Non-association has been reported between plasma t-PrP and the stage or disease duration, which argues against a potential use of this marker for prognostic purposes [96].

Advantages: It is possible to assess this biomarker in an accessible fluid, so invasive techniques are not required. The increase in t-PrP in the plasma of CJD (sporadic, acquired and genetic) can be detected from the early stages and does not vary between different stages of the disease, making it potentially useful as an early diagnosis biomarker.

Limitations: It is not specific for CJD diagnosis, and we do not have information about less prevalent prionopathies. This biomarker seems to be more related to neurodegeneration.

## 7. Nasal Mucosa

RT-QuIC performed in the samples obtained by nasal brushing, a less invasive technique compared to lumbar puncture, has shown a very high diagnosis accuracy according to preliminary data, with a sensitivity of 97% and specificity of 100%; this is a better performance compared to RT-QuIC in CSF [97,98]. It has been suggested that the combination of CSF RT-QuIC and RT-QuIC from a sample of nasal mucosa could increase the sensitivity in the early diagnosis of CJD to almost 100% [42,61].

Advantages: Obtaining a nasal mucosal sample would be safer and better tolerated by patients than lumbar puncture. If the preliminary data are confirmed, it could potentially replace CSF as a sample for RT-QuIC, or, at least, and this is ultimately more likely, allow increased diagnostic confidence in those subjects with high suspicion of prion disease with negative RT-QuIC in CSF, with the ability to detect some false negatives.

Limitations: Only preliminary data are available that do not allow evaluation of its practical clinical applicability. In many subjects, lumbar puncture will continue to be performed for screening of other neurodegenerative dementias; therefore, it seems implausible that nasal mucosa sampling can universally replace lumbar puncture for RT-QuIC study, at least not until other biomarkers with good diagnosis performance for other neurodegenerative dementias are developed for use in nasal mucosa.

## 8. Future Directions

The combination of neuroimaging (structural magnetic resonance) and CSF RT-QuIC for prion protein, t-tau, t-tau/p-tau ratio and 14-3-3 protein appears to be the most cost-effective combination for the early diagnosis of different prion diseases. Now that t-tau and t-tau/p-tau ratio are frequently assessed in mild cognitive impairment, considering whether abnormally high levels of one or both are highly suggestive of prion disease is essential. Qualitative analysis of EEG seems to be less useful for early diagnosis. However, research in the applicability of quantitative EEG to detect changes from earlier stages of the disease (including presymptomatic) could substantially increase its diagnostic performance, as well as reduce interobserver variability. PSG continues to be essential for the diagnosis of sFI and FFI, and review of EEG during PSG could potentially increase the capacity to detect PSWCs in CJD patients. Future studies to assess the performance of routine EEG vs. PSG-EEG (qualitative and quantitative analyses) would be of interest.

The performance of CSF biomarkers is very good, but their assessment still requires an invasive test (lumbar puncture) that is usually performed later after symptom onset. The identification of diagnostic biomarkers in more accessible fluids, such as peripheral blood, is currently being investigated. The most promising plasma biomarkers are very sensitive and non-specific (NfL and t-tau) and require ultra-sensitive analysis techniques that are unavailable in most centers. Nevertheless, despite technical limitations, peripheral blood is the best potential source for future biomarkers that, ideally, should also directly reflect prion pathology. PrPSc measured in exosomes, small extracellular vesicles capable of crossing the blood-brain barrier, in both peripheral blood and CSF, are being suggested as possible biomarkers for the accurate diagnosis of prion diseases, but validation studies are still lacking [99,100]. Even if the novel strategy of PrPSc determination in exosomes in peripheral blood fails, efforts to identify other biomarkers with the same, or at least very close, performance to RT-QuIC in CSF, the best available diagnostic biomarker, in more accessible fluids, such as peripheral blood and/or mucosa, will remain a priority.

Genetic prion diseases, on the other hand, are not only candidates for the application of potential future disease-modifying treatments (such as antisense oligonucleotides) but also allow the identification of prognostic and/or disease progression biomarkers. Confirming that periodic plasma NfL determination can predict up to two years in advance the onset of symptoms [90], and that higher t-tau levels are associated with a faster disease course, could help to better select potential candidates for future clinical trials. The availability of objective biomarkers of progression is essential to consider how to measure response in future clinical trials, as these are pathologies with widely varying clinical phenotypes and age of symptom onset [101]. Therefore, the identification of accessible prognostic biomarkers to be measured repeatedly in the population at risk is an undeniable need.

## 9. Conclusions

The early and accurate diagnosis of prion diseases with a clinically progressive course and without modifying treatment is essential. Therefore, the availability of biomarkers that can provide near-total diagnostic certainty is crucial. The fact that some of these biomarkers are found in high concentrations from an early stage (even in asymptomatic carriers in the genetic forms) can help us to better identify subjects that could be candidates for future therapeutic strategies.

Among the biomarkers available in clinical practice, RT-QuIC in cerebrospinal fluid, is the diagnostic biomarker with the best performance for early diagnosis; preliminary results of the same technique in other tissues, such as nasal mucosa, are also very promising. However, along with the determination of RT-QuIC, it is advisable to also request the determination of 14-3-3 protein, t-tau and p-tau, since diagnostic sensitivity will be further increased by their use without any decrease in specificity.

In recent years, there have been fewer advances in neuroimaging and neurophysiology biomarkers, but they are no less relevant. Structural neuroimaging (MR should be used preferably) within the etiological study of any neurodegenerative dementia is mandatory and will therefore continue to have a relevant role. Neurophysiological tests will continue to play an essential role in the diagnosis of sporadic or fatal familial insomnia, and the electroencephalogram, which is non-invasive and inexpensive, can remain as an additional diagnostic tool in centers with fewer resources.

For the time being, it does not seem that plasma biomarkers will be applied for the diagnosis of prion diseases due to the inferior performance of all biomarkers studied in the blood in comparison to cerebrospinal fluid to discriminate prion diseases from other neurodegenerative dementias. However, they could be useful for monitoring the response to a possible disease-modifying treatment, since assay protocols based on peripheral blood biomarkers as a target response will always be easier to perform than those based on CSF biomarkers.

## Figures and Tables

**Figure 1 medicina-58-00473-f001:**
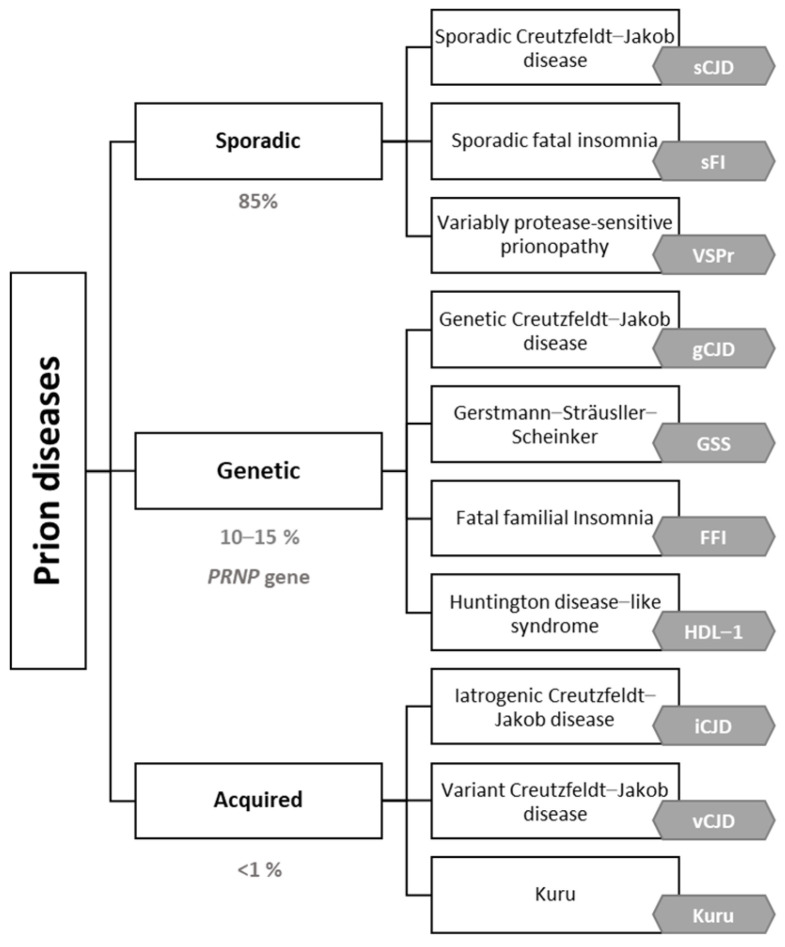
Prion diseases etiologic classification adapted by the authors [10,11,12,13].

**Figure 2 medicina-58-00473-f002:**
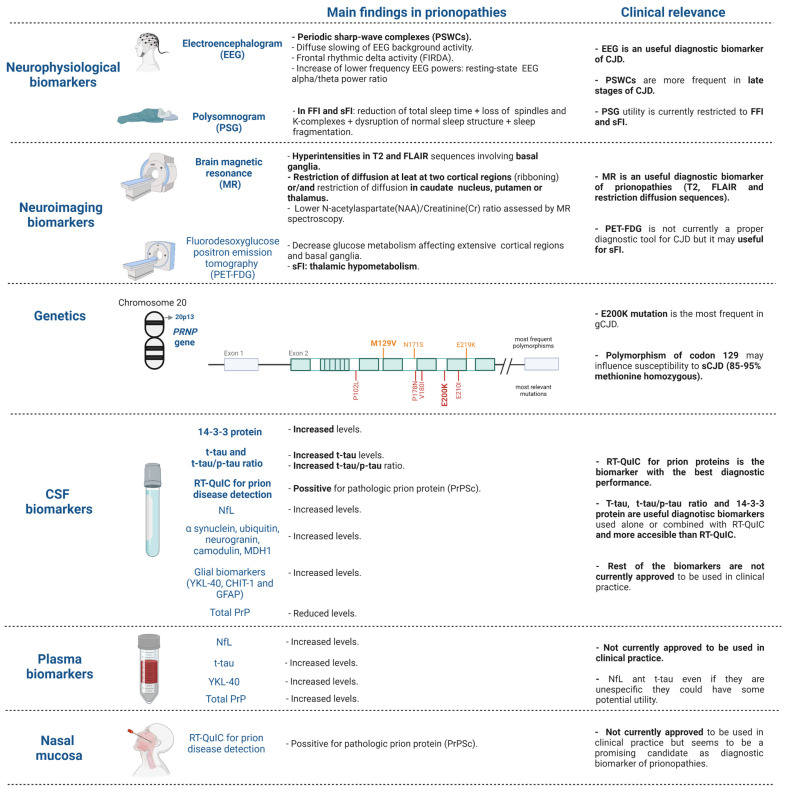
Diagnostic biomarkers for prion diseases in use in clinical practice and summary of the most promising biomarkers under investigation. Figure created with biorender.com (accessed on 18 February 2022). CJD: Creutzfeldt–Jakob diseases, sCJD: sporadic Creutzfeldt–Jakob disease; gCJD: genetic Creutzfeldt–Jakob disease, FFI: fatal familial insomnia, sFI: sporadic fatal insomnia.

**Figure 3 medicina-58-00473-f003:**
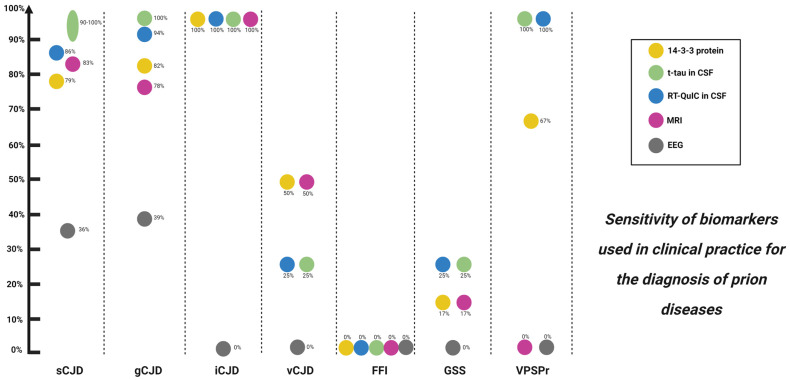
Biomarkers used in clinical practice and their diagnostic sensitivity for prion diseases. sCJD: sporadic Creutzfeldt–Jakob disease; gCJD: genetic Creutzfeldt–Jakob disease; iCJD: iatrogenic Creutzfeldt–Jakob disease; vCJD: variant Creutzfeldt–Jakob disease; FFI: fatal familial insomnia; GSS: Gerstmann–Sträusller–Scheinker; VPSPr: variably protease-sensitive prionopathy; MRI: magnetic resonance imaging; EEG: electroencephalogram.

**Figure 4 medicina-58-00473-f004:**
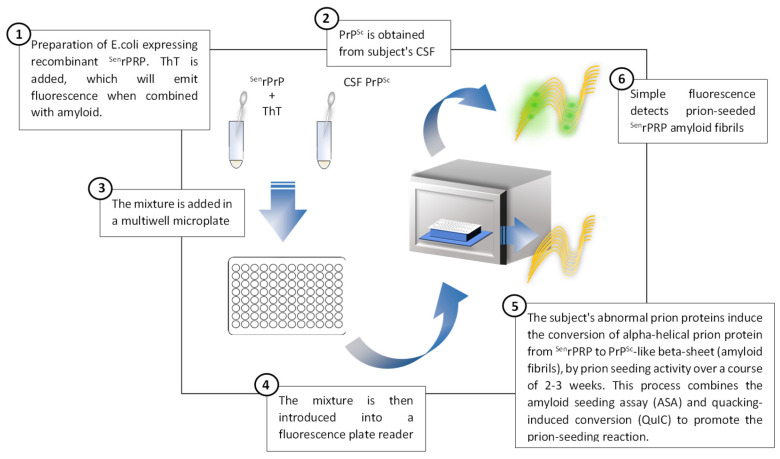
Description of the prion real-time quaking-induced conversion technique applied to the diagnosis of prion diseases, adapted from Orrù et al., 2017 [86].

**Table 1 medicina-58-00473-t001:** Diagnostic criteria for probable and definite categories of prion diseases.

		Probable Disease			Definitive Disease		
**CJD**	**sCJD**	Rapidly progressive cognitive impairment AND typical **EEG** (generalized periodic complexes) OR typical **MRI** (high signal in caudate/putamen or at least two cortical regions either on DWI or FLAIR) OR positive **14-3-3** AND two of the following: -Myoclonus -Visual or cerebellar problems -Pyramidal or extrapyramidal features -Akinetic mutism	OR	Progressive neurological syndrome AND positive **RT-QuIC** in CSF or other tissues.	Progressive neurological syndrome AND neuropathologically OR immunocytochemically OR biochemically confirmed.		
	**gCJD**	Progressive neuropsychiatric disorder AND definite or probable CJD in 1st degree relative	OR	Progressive neuropsychiatric disorder AND pathogenic *PRNP* mutation	Definitive CJD AND definitive or probable CJD in 1st degree relative	OR	Definitive CJD AND pathogenic *PRNP* mutation
	**vCJD**	Progressive neuropsychiatric disorder with duration of illness >6 months, no history of potential iatrogenic exposure, no evidence of gCJD and no alternative diagnosis suggested by routine investigations AND positive tonsil biopsy OR bilateral pulvinar high signal on MRI AND atypical appearance of sCJD on EEG in the early stages AND four of the following: -Early psychiatric symptoms (depression, anxiety, apathy, withdrawal, delusions) -Persistent painful sensory symptoms (frank pain or dysaesthesia) -Ataxia -Myoclonus, chorea or dystonia -Dementia			Progressive neuropsychiatric disorder AND neuropathological confirmation (spongiform change and extensive PrP deposition with florid plaques throughout the cerebrum and cerebellum)		
	**iCJD**	Progressive cerebellar syndrome in a recipient of human cadaveric-derived pituitary hormone OR sporadic CJD with a recognized exposure risk			Progressive cerebellar syndrome or sporadic CJD with a recognized exposure risk AND neuropathological confirmation		
**VSPr**		Cognitive impairment and/or two of the following: -Psychiatric symptoms -Parkinsonism -Aphasia -Ataxia -Myoclonus	AND	Less than 8 years duration AND absence of alternative etiology or phenotype divergence from atypical neurodegenerative dementias	Progressive neurological syndrome AND neuropathological confirmation		
**sFI and FFI**		Organic sleep disturbances. If not yet clinically apparent, a polysomnography has to be performed. At least two of the following: -Psychiatric symptoms (visual hallucinations, personality changes, depression, anxiety, aggressiveness, disinhibition, listlessness) -Ataxia -Visual symptoms -Myoclonus -Cognitive/mnesic deficits	AND	One of the following: -Loss of >10 kg during the last 6 months -Vegetative signs (hyperhidrosis, newly diagnosed arterial hypertonia, tachycardia, constipation, hyperthermia) -Husky voice	Progressive neurological syndrome AND neuropathological confirmation		
**GSS**		Progressive cerebellar syndrome, cognitive impairment and/or sensory symptoms AND pathogenic *PRNP* mutation			Progressive neurological syndrome AND neuropathological confirmation (amyloid deposits immunoreactive for PrP are the morphological hallmark of GSS)		
**HDL-1**		Abnormal involuntary movements, coordination difficulty, dementia, personality changes and psychiatric symptoms AND pathogenic *PRNP* mutation			Kuru and multicentric plaques that stain with anti-prion antibodies		

CJD: Creutzfeldt-Jakob disease; sCJD: sporadic Creutzfeldt-Jakob disease; EEG: electroencephalogram; MRI: magnetic resonance imaging; RT-QuIC: Real-Time Quacking-Induced conversion (RT-QuIC); CSF: cerebrospinal fluid; gCJD: genetic Creutzfeldt-Jakob disease; vCJD: variant Creutzfeldt-Jakob disease; iCJD: iatrogenic Creutzfeldt-Jakob disease; VSPr: Variably protease-sensitive prionopathy; sFI: sporadic fatal insomnia; FFI: fatal familial insomnia; GSS: Gerstmann-Sträusller-Scheinker; PrP: prion protein; HDL-1: Huntington disease-like syndrome. Bold words are for emphasis.

**Table 2 medicina-58-00473-t002:** Summary of the most frequent *PRNP* variants. Adapted from Ladogana et al., 2018 [2].

*PRNP* Variants	DNA Nucleotide Change	Predicted Protein Change	Related Prionopathy	Phenotype	Expected Survival (Median)
**P102L**	c.305C>T	Proline to leucine substitution at codon 102	**GSS**	**Ataxia** (100%), pyramidal (75%), dementia (62%), extrapyramidal (50%), myoclonus (25%). Others: dysarthria, sleep and sensory disturbances.	40 months.
**D178N**	c.532G>A	aspartic acid to asparagine substitution at codon 178	**FFI and genetic CJD** (Depends on the allele on codon 129. M allele: FFI, and V allele: genetic CJD)	**Dementia** (96%), myoclonus (89%), ataxia (82%), extrapyramidal (82%), pyramidal (79%), cortical blindness (79%). Others: sleep disturbances, dysarthria, weight loss and hyperhidrosis.	15 months (earlier onset and shorter duration of symptomatic disease in genetic CJD).
**V180I**	c.538G>A	Valine to isoleucine change at codon 180	**Genetic CJD**	**Dementia** (100%), extrapyramidal (54%), pyramidal (46%). Others: akinetic mutism (57%) and psychiatric (50%). Cortical blindness, myoclonus and ataxia are more infrequent.	16,4 months (wide range of survival).
**E200K**	c.598G>A	Glutamic acid to lysine substitution at codon 200	**Genetic CJD**	**Ataxia** (100%), dementia (95%), myoclonus (85%), pyramidal (70%), cortical blindness (70%), extrapyramidal (65%). Others: dysarthria, sleep disturbances and weight loss.	5 months (wide range of survival 1–74 months).
**V210I**	c.628G>A	Valine to isoleucine substitution at codon 210	**Genetic CJD**	**Ataxia** (100%), dementia (92%), myoclonus (92%), extrapyramidal (92%), cortical blindness (85%), pyramidal (72%). Others: dysarthria and sensory symptoms.	4 months (wide range of survival).

GSS: Gertsmann-Sträusller-Scheinker; FFI: Familial fatal insomnia; CJD: Creutzfeldt-Jakob disease; M: methionine; V: valine. Bold words are for emphasis.

## Data Availability

Not applicable.

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
