# Peer review of "Role of Biomarkers for the Diagnosis of Prion Diseases: A Narrative Review"

_medicina, 2022, doi:10.3390/medicina58040473_

Round 1
Reviewer 1 Report
In this manuscript, Altuna and colaborators review the latest investigation about diagnosis of prion diseases. The manuscript is well written and clear. I like how the authors classify the different techniques by subjects (neurophysiological markers, neuroimaging, CSF biomarkers...) and the part where authors write about advantages and limitations of each technique. I found this idea pretty useful. Howerve, there is some minor issues that should be attended before the article is published:
1- In line 38 the authors write: "...of a misfolded protein, scrapie (PrPSc), which propagates...". This is not the correct way to name the PrPSc. It should be called prion protein scrapie (PrPSc), as scrapie per se is not a protein, it is a disease. As this is the first time in the manuscript that PrP is named this should be better explained.
2- The Huntington disease-like 1 (HDL1) is a syndrome that usually is not classified as a prion disease. As you can see, it is never mentioned in any other article for the rest of the review. As I have seen, just the reference 6 talks directly about this disease and it is from 2001, what makes me think that probably this disease is an old fashioned way name for other prion disease (GSS or FFI). As you can read in the bibliography, prion diseases are highly variable and probably, this is an old naming for an odd subtype of other prion diease. Making it short, I would eliminate it from the review and I would not talk about it as, probably, is another naming for other prion disease. I think this is more a semantics problem than a real classifiable disease.
3- In line 49 there is a typo regarding the GSS spelling. It should be correctly noted as Gerstmann-Straüssler-Scheincker syndrome.
4- The part about CSF biomarkers - Biochemical analysis (lines 215 to 221) would be eliminated from the review as it is not giving any information. However, this is just a personal feeling and I understand if the authors want to maintain it.
5-In Figure 3 I do not understand the Genetics part (the squared part at the lowest left corner). I think this is not giving any important or useful information. Maybe some kind of graphic showing information about the different known mutations related to prion disease would be more interesting.
6-In Figure 4 there is a missunderstanding in plot 6. The authors write about "Simple immunofluorescence". There is no immunofluorescence as there is any antibody involved in this technique. The correct word to use should be "Simple fluorescence".
Author Response
In this manuscript, Altuna and colaborators review the latest investigation about diagnosis of prion diseases. The manuscript is well written and clear. I like how the authors classify the different techniques by subjects (neurophysiological markers, neuroimaging, CSF biomarkers...) and the part where authors write about advantages and limitations of each technique. I found this idea pretty useful. Howerve, there is some minor issues that should be attended before the article is published:
First of all thanks to the reviewer for his comments and suggestions for improvement of the article. We will answer each of your suggestions one by one.
1- In line 38 the authors write: "...of a misfolded protein, scrapie (PrPSc), which propagates...". This is not the correct way to name the PrPSc. It should be called prion protein scrapie (PrPSc), as scrapie per se is not a protein, it is a disease. As this is the first time in the manuscript that PrP is named this should be better explained.
There was indeed a typo that has been corrected and an explanatory sentence has been added following the reviewer's recommendations (it has been highlighted in red in the text itself along with the rest of the changes made).
2- The Huntington disease-like 1 (HDL1) is a syndrome that usually is not classified as a prion disease. As you can see, it is never mentioned in any other article for the rest of the review. As I have seen, just the reference 6 talks directly about this disease and it is from 2001, what makes me think that probably this disease is an old fashioned way name for other prion disease (GSS or FFI). As you can read in the bibliography, prion diseases are highly variable and probably, this is an old naming for an odd subtype of other prion diease. Making it short, I would eliminate it from the review and I would not talk about it as, probably, is another naming for other prion disease. I think this is more a semantics problem than a real classifiable disease.
We understand the reviewer's point of view. Our idea was that readers would identify the term Huntington disease-like 1 (HDL-1) as just another prion disease. We believe that it is somewhat unknown because of its rarity and that it is not reflected in the rest of the reviews on prion diseases. We would be in favor of keeping it as it is listed on the orphanet website as a prion disease updated in 2021 (https://www.orpha.net/consor/cgi-bin/OC_Exp.php?lng=EN&Expert=157941#:~:text=A%20rare%2C%20genetic%2C%20human%20prion,disorder%20and%20psychiatric%2Fbehavioral%20disturbances.) and in 2017 in OMIM (OMIM Entry - # 603218 - HUNTINGTON DISEASE-LIKE 1; HDL1).
3- In line 49 there is a typo regarding the GSS spelling. It should be correctly noted as Gerstmann-Straüssler-Scheincker syndrome.
The mistake has been corrected.
4- The part about CSF biomarkers - Biochemical analysis (lines 215 to 221) would be eliminated from the review as it is not giving any information. However, this is just a personal feeling and I understand if the authors want to maintain it.
We also debated whether or not to include it at the time of preparation. We do believe that it may be useful to keep it, although we understand that it is less informative than the rest of the sections referring to CSF. The idea we want to convey is that the cytobiochemical analysis is normal, unlike other causes with which it could potentially be confused, as in the case of autoimmune encephalitis.
5-In Figure 3 I do not understand the Genetics part (the squared part at the lowest left corner). I think this is not giving any important or useful information. Maybe some kind of graphic showing information about the different known mutations related to prion disease would be more interesting.
We have created another figure according to the suggestions of the other reviewer and subsection dedicated to genetics incorporating information about the most frequent mutations and polymorphisms in this new figure.
6-In Figure 4 there is a missunderstanding in plot 6. The authors write about "Simple immunofluorescence". There is no immunofluorescence as there is any antibody involved in this technique. The correct word to use should be "Simple fluorescence".
Thanks for your comment, the mistake has been corrected.

Reviewer 2 Report
- The title should mention the type. E.g. narrative review.
- Provide a reference for figure 1 classification.
- Table 1 should be edited.
- Table 2. Could the authors provide the significance of this table?
- Methods.
-The methods and flowchart can be provided as supplementary material since is a narrative review.
-Could the authors provide the full description of the mesh terms?
-Inclusion of articles besides authors' nationality language (French) should be provided some phrase regarding proficiency.
-What were the exclusion criteria? Did the authors find all the manuscripts full-text?
- Figure 3 is interesting, but the image should be of high quality. Also, it would be more interesting to do a figure of prion diseases and biomarkers without separating them in small tables.
- References should be updated. Only 40.44% comes from the last 5 years of science.
- The reviewer would like to ask the authors ‘‘what their study provides new for the literature?’’
------------------------------------------
New Ideas
Biomarkers
- A figure about all the biomarkers together would highly impact the quality of the manuscript. A figure different from figure 3 without scale; only biomarkers related to the sections of the manuscript.
Future directions
It is advised a chapter about what should future studies search?
Are the biomarkers that we have today enough? What should we develop? New biomarkers? Increase the specificity of the biomarkers already provided?
Could the mixture of biomarkers increase specificity? What is the best combination?
The description of the biomarkers helps the development of management.
Are biomarkers limited by geographical locations?
------------------------------------------
Thompson AGB, Mead SH. Review: Fluid biomarkers in the human prion diseases. Mol Cell Neurosci. 2019 Jun;97:81-92. doi: 10.1016/j.mcn.2018.12.003. Epub 2018 Dec 4. PMID: 30529227.
Vallabh SM, Minikel EV, Williams VJ, Carlyle BC, McManus AJ, Wennick CD, Bolling A, Trombetta BA, Urick D, Nobuhara CK, Gerber J, Duddy H, Lachmann I, Stehmann C, Collins SJ, Blennow K, Zetterberg H, Arnold SE. Cerebrospinal fluid and plasma biomarkers in individuals at risk for genetic prion disease. BMC Med. 2020 Jun 18;18(1):140. doi: 10.1186/s12916-020-01608-8. PMID: 32552681; PMCID: PMC7302371.
Mok TH, Mead S. Preclinical biomarkers of prion infection and neurodegeneration. Curr Opin Neurobiol. 2020 Apr;61:82-88. doi: 10.1016/j.conb.2020.01.009. Epub 2020 Feb 25. PMID: 32109717.
Author Response
First of all, thanks to the reviewer for his comments and suggestions for improvement of the article. We will answer each of your suggestions one by one.
- The title should mention the type. E.g. narrative review.
Following the reviewer's suggestions, the Narrative Review concept has been incorporated into the title (highlighted in red like the rest of the changes made).
- Provide a reference for figure 1 classification.
We have modified the caption of the figure. We have not really relied on a single source in the literature, but after reading multiple of the papers it is the adaptation made by the authors and that is why we had not added any specific reference. However, at the request of the reviewer, we have added the references of three works that have used the same classification scheme.
- Table 1 should be edited.
We have same done minor edition changes (removing lines mainly). We have incorporated to the manuscript as table format instead of picture format, so it should be easier to be further edited if required.
- Table 2. Could the authors provide the significance of this table?
We have added a commentary to the text. We do believe that this table provides information of interest to the potential reader because:
(a) The anatomoclinical correlate we believe is always of interest in any detailed description of a pathology or set of pathologies.
b) The post-mortem anatomopathological study is the way to definitively diagnose the existence or not of a prion disease and we believe that readers should be informed of the possible findings that can be found in this study and the relationship with the symptoms.
- Methods.
-The methods and flowchart can be provided as supplementary material since is a narrative review.
We accept the suggestion of the reviewer and we will move the Methods to the supplementary material.
-Could the authors provide the full description of the mesh terms?
We added the description note that is offered by Pubmed (a summary of it). As we comment we didn’t exclude any subheading.
-Inclusion of articles besides authors' nationality language (French) should be provided some phrase regarding proficiency.
We have added the next commentary: “In addition to the origin of the authors (two of them are also fluent in French as well as in Spanish and English), the decision to include Spanish and French was motivated by the fact that one of the highest incidence rates of prion diseases in Europe has been described in the Basque Country, located both in Spain and France.”
In the Basque Country is very frequent to learn almost at the same time basque, Spanish and French even earlier than English.
-What were the exclusion criteria? Did the authors find all the manuscripts full-text?
We added the requested information in the method section currently located in supplementary material. We acknowledge that the information is not as detailed as we would like (we do not note the exact number of articles we could not find but we know it is in the range of 5 to 10 having searched the repository of 2 universities, 3 research centers and 2 different hospitals) and that the exclusion criteria were rather soft. We believe, however, that such limitations are relatively common in narrative, not systematic, reviews such as the one we present.
- Figure 3 is interesting, but the image should be of high quality. Also, it would be more interesting to do a figure of prion diseases and biomarkers without separating them in small tables.
We have improved the quality of this figure (resolution) and modified the structure by removing the small tables.
- References should be updated. Only 40.44% comes from the last 5 years of science.
We have incorporated over 10 references from the last 5 years in comparison with the previous version of the manuscript.
- The reviewer would like to ask the authors ‘‘what their study provides new for the literature?’’
The combined examination of the current diagnostic criteria, of the multimodal diagnostic biomarkers (neurophysiological, neuroimaging, genetic, CSF and plasma analytes) both available for use in clinical practice and in the development phase, not only limited to CJD but also to the rest of the prionopathies, is one of the strengths of this work, not seen in most previous ones.
------------------------------------------
New Ideas
Biomarkers
- A figure about all the biomarkers together would highly impact the quality of the manuscript. A figure different from figure 3 without scale; only biomarkers related to the sections of the manuscript.
We have created another figure that that can be considered like an index of the biomarkers reviewed in the manuscript.
Future directions
We have incorporated a small section named future directions continuing with the suggestions of the reviewer and we have tried to answer the next questions.
It is advised a chapter about what should future studies search?
We have added a short comment about the necessity of future studies about prognostic biomarkers and we have introduce the research about extracellular vesicles both in plasma and cerebrospinal fluid.
Are the biomarkers that we have today enough? What should we develop? New biomarkers? Increase the specificity of the biomarkers already provided?
We believe that research in plasma biomarkers with high specificity for prion diseases is mandatory and we also believe that apart from CSF and plasma, research in quantitative assessment of neurophysiological tests could be of interest.
Could the mixture of biomarkers increase specificity? What is the best combination?
Information about the opinion of authors is now reflected also at the introduction of the short section of future directions as well as in the conclusion section.
The description of the biomarkers helps the development of management.
We have added a comment about the necessity of prognosis biomarkers in order to select the best candidates for clinical trials. We consider that this could be an example of how the selection of proper biomarkers can change the future management of the care of our patients.
Are biomarkers limited by geographical locations?
We consider that the most promising biomarkers can be use worldwide and changes on them are not so relevant to be commented in the text apart from the genetic section (we have added a short commentary).
We have checked that these 3 references are included in the manuscript.
Thompson AGB, Mead SH. Review: Fluid biomarkers in the human prion diseases. Mol Cell Neurosci. 2019 Jun;97:81-92. doi: 10.1016/j.mcn.2018.12.003. Epub 2018 Dec 4. PMID: 30529227.
Vallabh SM, Minikel EV, Williams VJ, Carlyle BC, McManus AJ, Wennick CD, Bolling A, Trombetta BA, Urick D, Nobuhara CK, Gerber J, Duddy H, Lachmann I, Stehmann C, Collins SJ, Blennow K, Zetterberg H, Arnold SE. Cerebrospinal fluid and plasma biomarkers in individuals at risk for genetic prion disease. BMC Med. 2020 Jun 18;18(1):140. doi: 10.1186/s12916-020-01608-8. PMID: 32552681; PMCID: PMC7302371.
Mok TH, Mead S. Preclinical biomarkers of prion infection and neurodegeneration. Curr Opin Neurobiol. 2020 Apr;61:82-88. doi: 10.1016/j.conb.2020.01.009. Epub 2020 Feb 25. PMID: 32109717.
Round 2
Reviewer 2 Report
- Table 1. Still should be edited, but the reviewers believe that the editorial process will assist that.
- Table 2. What is the significance of this table regarding the aim of this review?
- Even though the authors did not exclude any of the subheadings. It is advised to provide a full description of the mesh term. E.g.
Search: Tremor
Mesh terms: "tremor"[MeSH Terms] OR "tremor"[All Fields] OR "tremors"[All Fields] OR "tremoring"[All Fields] OR "tremorous"[All Fields]
Author Response
First of all we would like to thank the reviewer for his contribution to improve our work. We have responded point by point to his comments.
- Table 1. Still should be edited, but the reviewers believe that the editorial process will assist that.
We would appreciate any help in editing the table so that it complies with a certain format that is easier to read and/or better suits the aesthetics of the journal. We are always eager to learn how to improve the communication of our results.
2. Table 2. What is the significance of this table regarding the aim of this review?
The authors believed that through this table we were highlighting that even in the age of biomarkers anatomopathological studies still has relevance. However, understanding that this is the second time we have been asked about the appropriateness of this table for this review, we have decided to remove it following the reviewer's suggestions.
3. Even though the authors did not exclude any of the subheadings. It is advised to provide a full description of the mesh term. E.g.
The search strategy used in the Pubmed Search Builder was: "Prion Diseases" [Mesh] AND "Biomarkers" [Mesh]. We have modified by adding this information in the supplementary material section. As indicated a posteriori we added other articles not initially identified when reviewing the bibliography of the initially identified articles.